# Treg Therapy for the Induction of Immune Tolerance in Transplantation—Not Lost in Translation?

**DOI:** 10.3390/ijms24021752

**Published:** 2023-01-16

**Authors:** Nina Pilat, Romy Steiner, Jonathan Sprent

**Affiliations:** 1Department of Cardiac Surgery, Medical University of Vienna, 1090 Vienna, Austria; 2Center for Biomedical Research, Medical University of Vienna, 1090 Vienna, Austria; 3Immunology Division, Garvan Institute of Medical Research, Darlinghurst, NSW 2010, Australia; 4St Vincent’s Clinical School, University of New South Wales, Sydney, NSW 2010, Australia

**Keywords:** transplantation, regulatory T cells, tolerance

## Abstract

The clinical success of solid organ transplantation is still limited by the insufficiency of immunosuppressive regimens to control chronic rejection and late graft loss. Moreover, serious side effects caused by chronic immunosuppressive treatment increase morbidity and mortality in transplant patients. Regulatory T cells (Tregs) have proven to be efficient in the induction of allograft tolerance and prolongation of graft survival in numerous preclinical models, and treatment has now moved to the clinics. The results of the first Treg-based clinical trials seem promising, proving the feasibility and safety of Treg therapy in clinical organ transplantation. However, many questions regarding Treg phenotype, optimum dosage, antigen-specificity, adjunct immunosuppressants and efficacy remain open. This review summarizes the results of the first Treg-based clinical trials for tolerance induction in solid organ transplantation and recapitulates what we have learnt so far and which questions need to be resolved before Treg therapy can become part of daily clinical practice. In addition, we discuss new strategies being developed for induction of donor-specific tolerance in solid organ transplantation with the clinical aims of prolonged graft survival and minimization of immunosuppression.

## 1. Introduction

Clinical solid organ transplantation (SOT) is the only curative treatment for various end-stage organ diseases. However, long-term graft survival is limited by incomplete control of the immune reaction to the donor alloantigens that lead to graft rejection. Immunosuppressive drugs clearly improve short-term graft survival, but they are often unable to control chronic rejection. Moreover, nonspecific chronic immunosuppression leads to increased morbidity and mortality amongst transplant recipients.

Inducing donor-specific transplantation tolerance in the absence of chronic immunosuppression has been a major clinical goal since the original description of acquired tolerance induction in mice by Medawar and colleagues in the 1950s [1]. However, despite impressive progress in multiple rodent models, no tolerance protocol has proved robust and safe enough for widespread clinical use [2]. In recent years, there has been much interest in the use of regulatory T cells (Tregs) for tolerance induction. Tregs are a specialized subset of CD4 T cells characterized by high and stable intracellular expression of the transcription factor FoxP3, high surface expression of the high-affinity IL-2 receptor CD25 and low expression of IL-7 receptor CD127 [3]. Since Foxp3 deficiency leads to lethal autoimmunity, it is well established that Tregs play a crucial role in self-tolerance. Nevertheless, not all cells with Treg function are Foxp3+. Thus, atypical Foxp3- Tregs with various phenotypes, including CD4+CD25−CD69+LAP+ type 3 helper cells (Th3), CD4+IL-10+ (Tr1), CD8+CD28- and certain CD4−CD8- double-negative cells, can display potent immunosuppressive activity, mainly by the production of anti-inflammatory cytokines [4]. Typical Foxp3+ Tregs originate in the thymus by positive selection in the cortex through their high-affinity T-cell receptor (TCR) interaction with self-peptides presented by thymic stromal cells; these Tregs are referred to as thymic Tregs (tTregs) or natural Tregs (nTregs). Under inflammatory conditions, however, conventional T cells can differentiate into a subset of Foxp3+ Tregs in the post-thymic environment; these peripheral Tregs (pTregs) may differ from tTregs in their suppressive functions and also in their TCR repertoire. Upregulation of Foxp3 by pTregs correlates with their distinct demethylation pattern of the TDSR region. Thus, demethylation of this enhancer region within the FoxP3 gene is necessary for stable expression of this transcription factor by pTregs, with incomplete demethylation risking redifferentiation into effector T cells [5].

Distinguishing phenotypically between tTregs and pTregs is currently difficult. Both subsets express helios [6], although some pTregs differ from tTregs in showing low expressions of neuropilin-1 (Nrp-1) [7]. However, this difference applies only to circulating cells, as pTregs have been shown to upregulate Nrp-1 expression during inflammation [8]. Although there is still debate on the phenotype of tTregs and pTregs [9], recent studies on genetically modified mouse strains that favor pTreg or tTreg formation have failed to reliably distinguish between these cells on the basis of helios and neuropilin-1 [9]. Nevertheless, pTregs and tTregs do differ in their stability of Foxp3 expression. Thus, unlike tTregs, pTregs may lose FoxP3 expression under inflammatory conditions and differentiate into a Th17-like phenotype [10].

Recently, on the basis of their patterns of differentiation, three subpopulations of CD4+FoxP3+ Tregs have been described in both mice and humans, namely central Tregs (cTreg), displaying a naive-like or resting phenotype; effector Tregs (eTreg), defined by an activated phenotype; and memory Tregs (mTreg), showing a long lifespan and the phenotype of memory T cells. These subsets can be further subdivided into central memory (cmTregs) and effector memory (em Tregs) Tregs (Figure 1) and are reported to differ with regard to their functions in vivo and also in the expression of surface molecules (chemokine receptors and adhesion molecules) and transcriptional patterns [11].

In this review, we will focus on the therapeutic potential of prototypical CD4+FoxP3+ Tregs and the preliminary results of Treg cell-based therapies in clinical trials. As shown in animal models, adoptive transfer of Tregs is an attractive therapeutic approach for restoring self tolerance in autoimmune diseases and also for the induction of specific tolerance towards allogeneic organ allografts. Clinically, Treg therapy has proven effective in the prevention/treatment of graft-versus-host disease (GvHD). In addition, clinical trials have demonstrated the feasibility and safety of Treg cell-based therapies for autoimmune diseases, hematopoietic stem cell transplantation and SOT. However, these results are still very preliminary. Thus, whereas results in type 1 diabetes (T1D) and GvHD are promising [12], only one SOT study proving efficacy has been published so far; this study was designed specifically for weaning the patients off immunosuppression (see below [13]).

Below, we discuss the published and ongoing clinical trials that validate the efficacy and therapeutic potential of Treg infusions in SOT. We also review improvements in understanding some of the many open questions on Treg cell-based therapies with regard to cell source, cell numbers, lineage stability, specificity, lack of biomarkers, functional assays or other valid surrogate endpoints for evaluating efficacy in clinical trials.

## 2. Current Status and Ongoing Studies

More than 10 years after the first-in-man trial using Treg adoptive cell therapy in patients with GvHD [14], valuable information has arisen from multiple phase I to phase I/II clinical trials designed to test the safety, feasibility and efficacy of Treg therapy in solid organ transplantation (Table 1). Until now, only a few reports have been published and data on efficacy are still scarce.

An important study with regard to efficacy was published in 2016 by Todo et al. [13]. Notably, these workers were able to induce operational tolerance with a single infusion of a non-GMP-compliant cell product enriched for donor-specific Tregs in 7 out of 10 (splenectomized) living-donor liver transplant recipients. This is the first study to demonstrate successful discontinuation of immunosuppressive medication following Treg-based therapy. Here, recipient lymphocytes and splenocytes were cocultured in vitro with irradiated donor splenocytes in combination with anti-CD80/CD86 antibodies for 2 weeks in order to obtain a final cell product that included Tregs at 0.43–6.37 × 10^6^/kg. However, this approach was only successful in transplant patients devoid of other immunological disorders such as autoimmune diseases. It should be mentioned that in this study, only 3–17% of the cell product waa defined as Tregs, thus making it difficult to determine the immunoregulatory mechanisms involved [13].

One year later, Chandran et al. at UCSF, USA, published a report demonstrating the safety and feasibility of transferring FACS-sorted, autologous CD4+CD127lo/-CD25+ Tregs back into patients with subclinical kidney graft inflammation, detected by surveillance biopsy during the 6-month post-transplant period. The infused cells were polyclonally expanded in vitro for 14 days before injection and then transferred in a single dose of 320 × 10^6^ Tregs/patient; the recipients were maintained on immunosuppression with tacrolimus, mycophenolate mofetil and prednisone. No serious side effects were seen and the authors were able to track the infused Treg cells in vivo for up to 1 month after transfer by in vitro deuterated glucose labeling [15]. Though based on studies with only three patients, the data are promising, and studies with larger patient numbers are in progress (NCT02711826).

In 2018, Mathew et al. from Northwestern University, USA, published the results of the TRACT trial, a study using a polyclonal Treg product for tolerance induction in de novo kidney transplant patients. In this study, autologous, polyclonally expanded CD4+ CD25+ Tregs were isolated from leukapheresis products using the CliniMACS system; then, single doses of 0.5, 1 or 5 × 10^9^ cells were administered 60 days post-kidney transplantation. In addition to Treg cell infusion, patients received tacrolimus, mycophenolate and corticosteroid induction therapy with lymphodepletion (alemtuzumab) 2 days before transplantation; the protocol included conversion from tacrolimus to sirolimus, which was conducted one month post-transplantation. During the follow-up period, no cell therapy-related severe adverse events were reported. However, there was an increase in opportunistic infections and de novo DSA development (two of nine patients). Whether these findings were related to the infusion of Tregs or suboptimal immunosuppression was unclear [16].

Between 2012 and 2018, the ONE study aimed to develop a unified approach for the evaluation of cellular immunotherapy in solid organ transplantation. This multicenter trial involved eight academic institutions in Europe and the USA that were testing the safety and feasibility of six different regulatory cell subsets (four Treg cell products) approved for manufacture and therapeutic testing in de novo kidney transplant patients. The goal was a direct comparison of the safety, clinical practicality and therapeutic efficacy of different types of regulatory cells with the ultimate goal of preventing immunological rejection without the need for chronic pharmacological immunosuppression. For the study by King’s College and the University of Oxford, UK, the investigators tested a polyclonally expanded Treg product (pTreg1 [17]) isolated from the peripheral blood using the CliniMACS system. Four different Treg concentrations (1 × 10^6^; 3 × 10^6^; 6 × 10^6^; 10 × 10^6^) were tested as a single dosage on day 5 post-transplantation in a total of 15 patients enrolled (NCT02129881). At the Charite University of Berlin, Germany, another polyclonal Treg product (pTreg 2 [18]) was approved for manufacturing and use in kidney transplant recipients. As in the Oxford study, polyclonally expanded Tregs were used at different therapeutic doses and were infused at day 7 post-transplantation in 17 enrolled patients (NCT02371434) (San Francisco; NCT02371434). The other two Treg cell products were tested at the University of California (UCSF) (NCT02244801) and Massachusetts General Hospital (Boston, MA; NCT02091232), USA, and were generated in the presence of donor PBMCs to enhance their specificity and efficacy. In these studies, the UCSF group made Tregs with donor specificity by coculturing Tregs with hCD40L-expressing K562-activated donor B cells restimulated with CD3/CD28-coated beads (donor antigen reactive (dar) Treg product; darTreg-sBC; San Francisco, CA, USA) [19], whereas the Boston group cultivated recipient PBMC-derived Tregs with irradiated donor PBMCs under the cover of costimulation blockade with belatacept (darTreg-CSB). In both trials, up to 9 × 10^8^ darTregs were infused as a single dosage on day 10 after living kidney transplantation [20].

Collectively, these studies established the safety and feasibility of use for each of the cell products tested; notably, fewer incidences of opportunistic infections were seen in the cell therapy groups (CTG) than the reference group trial (RGT). Efficacy was evaluated for a combined ONE study CTG group (including two trials using monocyte-derived cell products) in the first report, with 40% (15/38) of patients successfully switching to Tacrolimus monotherapy [21]. Following the results from the ONE study, the phase IIb trial (TWO Study) started recruiting in 2019 and will further evaluate polyclonal Treg therapy for their efficacy to allow for a reduction of immunosuppressive medication in renal transplantation patients. When available, details of the individual trial arms will hopefully provide important additional insights into the feasibility, safety and efficacy of each Treg cell therapy product.

Despite the promise of the above studies, there were some potential concerns. Thus, one patient in the darTreg-sBC trial developed acute signs of chronic rejection, thereby resulting in termination of this trial arm (https://cordis.europa.eu/project/id/260687/reporting (accessed on 26 September 2022)). In addition, in liver transplantation patients, manufacturing problems encountered in 5 out of 10 patients during preparation of donor-specific Tregs using donor B-cells led to termination of the deLTa trial (NCT02188719). With regard to efficacy, it should also be mentioned that attempts to withdraw immunosuppression from patients who received the autologous, donor-specific Treg product led to the development of rejection episodes in five patients (NCT02474199). Hopefully, these problems will be resolved in future studies [22].

**Table 1 ijms-24-01752-t001:** Ongoing clinical trials adopting regulatory T-cell therapy in solid organ transplantation (search date 26 September 2022).

Study ID		Age	Title	Product	Dose	Status	Location	Aim/Results
Renal Transplantation—endogenous Treg expansion								
**NCT02417870**	I/II	18–75 (adult, older adult)	Ultra-low Dose Subcutaneous IL-2 in Renal Transplantation	Low-dose recombinant IL-2 (proleukin)	-	Terminated (June 2021)	Brigham and Women’s Hospital, Boston, MA, US	Safety and efficacy of treatment with low-dose rIL-2 in renal transplant recipients.
**Renal Transplantation—adoptive Treg therapy**								
**NCT02088931**	I	18–50 (adult)	Treg Adoptive Therapy for Subclinical Inflammation in Kidney Transplantation (TASK)	CD4+CD127lo/-CD25+ polyclonally expanded Tregs	3.2 × 10^8^	Completed (July 2022)	University of California, San Francisco, CA, US	**Results: Approach is safe and feasible. One patient developed acute cellular rejection. Infused Tregs remained detectable for 1 month.** [15]
**NCT02091232**	I	>18 (adult, older adult)	Infusion of T-Regulatory Cells in Kidney Transplant Recipients (The ONE Study)	Tregs (recipient) stimulated with donor PBMCs and belatacept	4–9 × 10^8^	Completed (Nov 2021)	Massachusetts General Hospital, Boston, MA, US	To examine in living donor renal transplant recipients the safety and feasibility of administering T regulatory cells derived from recipient PBMC stimulated with kidney donor PBMC in the presence of costimulatory blockade with belatacept. [21]
**NCT04817774**	I/II	18–70 (adult, older adult)	Safety & Tolerability Study of Chimeric Antigen Receptor T-Reg Cell Therapy in Living Donor Renal Transplant Recipients	CD4+ CD45RA+ CD25+ CD127low/- HLA-A*02-specific CAR Tregs	-	Recruiting (Dec 2021)	University Hospitals Leuven, Leuven, Belgium (and 3 other centers)	A multicenter, first-in-human, open-label, single-ascending-dose, dose-ranging study of autologous, chimeric antigen receptor T regulatory cells (CAR-Treg) in HLA-A2-mismatched living-donor kidney transplant recipients.
**NCT03943238**	I	18–65 (adult, older adult)	TLI, TBI, ATG & Hematopoietic Stem Cell Transplantation and Recipient T Regs Therapy in Living Donor Kidney Transplantation	Autologous polyclonally expanded Tregs	Starting at 25 × 10^6^/kg	Recruiting (May 2022)	Stanford University, Palo Alto, Northwestern University, Chicago, IL, US	To determine if total lymphoid irradiation (TLI) in combination with anti-thymocyte globulin (ATG) and infusion of donor hematopoietic stem cells along with recipient Tregs will allow for discontinuation of immunosuppressive treatment after living-donor kidney transplantation.
**NCT03284242**	n/a	18–65 (adult, older adult)	A Pilot Study Using Autologous Regulatory T Cell Infusion Zortress (Everolimus) in Renal Transplant Recipients	Autologous polyclonally expanded Tregs	n/a	Recruiting (May 2022)	University of Kentucky Medical Center, Lexington, Kentucky, US	Safety and effectiveness of infusion of autologous polyclonally expanded Tregs to renal transplant recipients receiving Zortress (Everolimus) as immunosuppressive therapy.
**NCT02711826**	I/II	>18 (adult, older adult)	Treg Therapy in Subclinical Inflammation in Kidney Transplantation	Autologous polyclonally expanded Tregs	5.5 ± 4.5 × 10^8^	Recruiting (March 2022)	University of California at San Francisco, CA, US (and 5 other centers)	To determine the safety and efficacy of a single dose of autologous polyclonal Tregs in renal transplant recipients with subclinical inflammation (SCI) in the 3–7 months post-transplant allograft protocol biopsy compared to control patients treated with CNI-based immunosuppression.
**NCT02145325**	I	18–65 (adult, older adult)	Trial of Adoptive Immunotherapy with TRACT to Prevent Rejection in Living Donor Kidney Transplant Recipients	Autologous polyclonal expanded CD4+CD25+ nTregs	0.5–5 × 10^6^	Completed (Oct 2019)	Northwestern University Comprehensive Transplant Center, Chicago, Illinois, US	**Results: Approach is safe and feasible. Circulating levels of Tregs were increased for 1-year follow-up.** [16]
**NCT03867617**	I/II	>18 (adult, older adult)	Cell Therapy for Immunomodulation in Kidney Transplantation	Autologous polyclonally expanded CD4+ CD127lo/- CD25+ CD45RA Tregs	0.3–1.5 × 10^7^	Recruiting (Sep 2019)	Medical University of Vienna, Vienna, Austria	Treatment combining ex vivo expanded recipient regulatory T cells with donor bone marrow and Tocilizumab for feasible, safe and efficacious induction of transient chimerism in living-donor kidney transplant recipients.
**NCT01446484**	I/II	1–18 (child)	Treatment of Children with Kidney Transplants by Injection of CD4+CD25+FoxP3+ T Cells to Prevent Organ Rejection	Autologous CD4+ CD25+ CD127low FoxP3+ Tregs	2 × 10^8^	Unknown (Nov 2011)	Russian State Medical University, Moscow, Russian Federation	This study will evaluate the treatment of children who received a kidney transplant with Alemtuzumab or other immunosuppressing medications in combination with injection of autologous ex vivo expanded Tregs.
**NCT02371434**	I/II	18–65 (adult, older adult)	The ONE Study nTreg Trial (ONEnTreg13)	Autologous polyclonally expanded CD4+ CD25+ FoxP3+ nTregs	0.5–3 × 10^6^	Completed (Feb 2020)	Charité University Medicine, Berlin, Germany	**Results: Tapering of immunosuppressive medication to low-dose tacrolimus is safe and feasible. (n = 8 patients)** [21]
**NCT02244801**	I	18–70 (adult, older adult)	Donor-Alloantigen-Reactive Regulatory T Cell (darTreg) Therapy in Renal Transplantation (The ONE Study)	Donor-alloantigen-reactive Tregs (darTregs)	3 × 10^8^; 9 × 10^8^	Completed (Oct 2018)	University of California San Francisco, CA, US	**Results: Treg-based therapy is achievable and safe in living-donor kidney transplant recipients. This approach is associated with fewer infectious complications, but similar rejection rates in the first year. darTregs: Tregs stimulated with B cells that had been activated with K562 cells expressing hCD40L.** [21]
**NCT02129881**	I/II	>18 (adult, older adult)	The ONE Study UK Treg Trial	Autologous polyclonally expanded Tregs	1–10 × 10^6^/kg	Completed (Jan 2019)	Guy’s Hospital, London, UK	**Results: Approach is safe and feasible. Mycophenolate Mofetil (MMF) withdrawn and on Tacrolimus monotherapy. (n = 4 patients) Less opportunistic infections and transient increase of Treg cell numbers were detected.** [21]
**ISRCTN** **11038572**	II	>18 (adult, older adult)	TWO study: cell therapy trial in renal transplantation	Autologous polyclonally expanded Tregs	5–10 × 10^6^/kg	Recruiting (June 2022)	Oxford Transplant Centre, Churchill Hospital, Oxford, UK	This study aims to demonstrate the efficacy of polyclonal Tregs with the goal of allowing for reduction in immunosuppression to a single drug by 6 months post-transplantation. [23]
**Liver Transplantation—endogenous Treg expansion**								
**NCT02739412**	II	18–65 (adult, older adult)	Efficacy of Low Dose, SubQ Interleukin-2 (IL-2) to Expand Endogenous Regulatory T-Cells in Liver Transplant Recipients	Low-dose recombinant IL-2 (proleukin)	0.30MIU/m^2^ body surface area; for 4 weeks	Active, not recruiting (May 2021)	Beth Israel Deaconess Medical Center, Boston, MA, US	Aim of this study is to investigate if very-low-dose IL-2, given to liver transplant patients by subcutaneous injections, over a 4-week period of time, will cause an increase in the number of Treg cells in the blood. Includes analysis regarding safety of treatment.
**NCT02949492**	IV	18–50 (adult)	Low-dose IL-2 for Treg Expansion and Tolerance (LITE)	Low-dose recombinant IL-2 (proleukin)	-	Terminated (Aug 2019)	King’s College Hospital London, UK	**Results: Stable patients 2–6 years post-liver transplantation were treated with low-dose IL-2 to facilitate discontinuation of immunosuppression. Patients achieved a 2-fold increase in circulating Tregs; the trial was terminated after 6 patients developed rejection requiring immunosuppression reinstitution.** [24]
**Liver Transplantation—adoptive cell therapy**								
**NCT01624077**	I	10–60 (child, adult)	Safety Study of Using Regulatory T Cells Induce Liver Transplantation Tolerance	Autologous polyclonally TGF-β induced CD4+ CD25+ CD127- Tregs	1 × 10^6^/kg	Unknown (Feb 2015)	Nanjing Medical University, Nanjing, Jiangsu, China	Generation of donor-alloantigen-specific CD4+CD25+ Tregs from peripheral blood of pretransplant patients, for graft-specific tolerance induction.
**NCT03654040**	I/II	18–70 (adult, older adult)	Liver Transplantation with Tregs at UCSF	Autologous expanded donor-alloantigen-reactive Tregs (arTregs)	30–90 × 10^6^ total Treg cells	Recruiting (Aug 2021)	University of California, San Francisco, San Francisco, California, US	A single-center, prospective, open-label, nonrandomized clinical trial using alloantigen-specific Tregs to facilitate immunosuppression withdrawal in liver transplant recipients.
**NCT03577431**	I/II	18–70 (adult, older adult)	Liver Transplantation with Tregs at MGH	Autologous expanded donor-alloantigen-reactive CD4+ CD25+ CD127- Treg cells (arTregs)	2.5–125 × 10^6^	Recruiting (Nov 2021)	Massachusetts General Hospital: Transplantation, Boston, MA, United States	A single-center, prospective, open-label, nonrandomized clinical trial exploring cellular therapy to facilitate immunosuppression withdrawal in liver transplant recipients.
**NCT02474199**	I/II	18–70 (adult, older adult)	Donor Alloantigen Reactive Tregs (darTregs) for Calcineurin Inhibitor (CNI) Reduction (ARTEMIS)	Autologous donor-alloantigen-reactive Tregs (darTregs)	3–5 × 10^8^	Completed (Feb 2021)	University of California at San Francisco, San Francisco, CA, US Northwestern University Comprehensive Transplant Ctr, Chicago, IL, US, Mayo Clinic in Rochester, Rochester, NY, US	Safety of donor-alloantigen-reactive Tregs to facilitate minimization and/or discontinuation of immunosuppression in adult liver transplant recipients. **Results: Problems with Treg product manufacturing; discontinuation of immunosuppression not possible.**
**NCT02188719**	I	21–70 (adult, older adult)	Donor-Alloantigen-Reactive Regulatory T Cell (darTregs) in Liver Transplantation (deLTa)	Autologous donor-alloantigen-reactive Tregs (darTregs)	2.5–96 × 10^7^	Terminated (Sep 2020)—has results	University of California at San Francisco, San Francisco, CA, US; Northwestern University Comprehensive Transplant Ctr, Chicago, IL, US, Mayo Clinic in Rochester, Rochester, NY, US	Safety of receiving one or three different doses of donor-alloantigen-reactive Tregs (darTregs) while receiving a specific drug combination. **Results: issues regarding donor-specific Treg manufacturing using donor B-cells led to termination.**
**NCT02166177**	I/II	18–70 (adult, older adult)	Safety and Efficacy Study of Regulatory T Cell Therapy in Liver Transplant Patients (ThRIL)	Autologous polyclonally expanded Tregs	0.5–1; 3–4.5 × 10^6^/kg	Completed (Jan 2019)	King’s College Hospital, London, UK	**Results: Safety of Treg transfer was confirmed. Transient increase of the pool of circulating Tregs and reduced anti-donor T-cell responses were detected. Low applicability of earlier Treg dose (3 months post-transplant).** [25]
**UMIN-000015789**	I/II	18–65 (adult, older adult)	Tolerance induction by a regulatory T cell-based cell therapy in living donor liver transplantation	Donor-reactive Treg-enriched cell product	0.23–6.37 × 10^6^ Tregs/kg	Recruiting (until July 2012) Data published 2016	Hokkaidou University Graduate School of Medicine, Japan	**Results: 7 of 10 patients are immunosuppressant-free for >6 years.** [13]
**NCT05234190**	I/II	18–70 (adult, older adult)	Safety and Clinical Activity of QEL-001 in A2-mismatch Liver Transplant Patients (LIBERATE)	Autologous CAR Tregs targeting HLA-A2 (HLA-A2 CAR-Treg)	-	Recruiting (Feb 2022)	Cambridge University Hospitals NHS Foundation Trust, Cambridge, UK Royal Free London NHS Foundation Trust, London, UK King’s College Hospital NHS Foundation Trust London, UK	A multicenter, first-in-human, open-label, single-arm study of an autologous CAR T regulatory (CAR-Treg) in HLA-A2-mismatched liver transplant recipients. The aim is for the CAR-Tregs to be activated on recognition of HLA-A2 antigens present on the donated liver and subsequently induce and maintain immunological tolerance to the organ.
**Heart Transplantation—adoptive cell therapy**								
**NCT04924491**	I/II	0–2 (child)	Cell Therapy with Treg Cells Obtained from Thymic Tissue (thyTreg) to Prevent Rejection in Heart Transplant Children (THYTECH)	Autologous thyTreg	10–20 × 10^6^ thyTreg /kg	Recruiting (Aug 2022)	Hospital General Universitario Gregorio Marañon Madrid, Spain	A phase I/II clinical trial testing the safety and efficacy of the adoptive transfer of autologous Treg cells from thymic tissue (thyTreg) discarded in pediatric cardiac surgeries to prevent rejection in heart transplant children.

As emphasized earlier, for organ transplantation, the overall goal of Treg therapy is to abolish or reduce the need for chronic immunosuppression. To date, however, the results have been only mildly encouraging. For liver transplantation, therapy with polyclonally expanded Tregs has been tested in several clinical trials, thus far limited to patients at 6–12 months post-transplant [25]. However, with the exception of the study conducted by Todo et al. [13], freedom from immunosuppression therapy was not achieved (NCT02166177). Nevertheless, Treg therapy was shown to be safe and led to a transient increase in circulating Tregs and reduced anti-donor T-cell responses. Here, in a recent report, low-dose IL-2 treatment was used to try to increase Treg survival in stable-liver recipients tested at 2–6 years post-transplantation, with the aim of complete discontinuation of immunosuppression (NCT02949492). Unexpectedly, however, although levels of Tregs were increased in the blood, intrahepatic expansion of Tregs was not seen. After initiation of weaning, all patients failed to reach the first endpoint due to rejection and requirement of immunosuppression restoration, resulting in termination of the trial [24].

Despite the capability of Tregs to impede graft rejection in preclinical studies, we are still far from achieving this goal in the clinic. As described above, comparing the results achieved to date on Treg therapy is very difficult because the Treg populations studied differed by multiple parameters, including generation, purity, phenotype, dose and time of injection. Hopefully, further studies will eventually lead to selection of a “standard” population of Tregs for clinical use. Real success here is far from guaranteed, however, and may hinge on the development of new and safe techniques for promoting Treg survival and/or the creation of fundamentally new types of Tregs, notably chimeric antigen receptor (CAR) Tregs (see below; NCT05234190).

## 3. Open Questions

### 3.1. Specificity and Efficacy

Most preclinical studies as well as clinical trials have involved the adoptive transfer of polyclonal Tregs comprising multiple TCR specificities. These Treg populations regulate effector T cells through antigen-independent “bystander” suppression and are usually generated by expansion in vitro for several days by polyclonal stimulation via CD3 and CD28 (mostly coated on plates or expander beads) (Figure 2). The alternative approach is to select Tregs for their TCR specificity, e.g., for the donor alloantigens in the case of SOT. Here, numerous preclinical studies have suggested that antigen-specific Tregs are indeed more efficient than polyclonal Tregs in the suppression of allograft rejection, largely due to their improved capacity to home to the target organ concerned. Antigen-specific Tregs can be selectively enriched by in vitro coculture with donor cells/APCs pulsed with donor antigen. Here, in a recent study, investigators directly compared antigen-specific Tregs that were expanded ex vivo using allogeneic stimulation by either B cells or DCs. Both darTreg products showed comparable levels of FOXP3, HELIOS, CD25, CD27 and CD62L, demethylated FOXP3 enhancer and in vitro suppressive function. Notably, DC-darTregs were generated in 2-fold higher numbers, suggesting that DC are better than B cells for expansion [26].

An alternative to preparing antigen-specific Tregs is to transduce normal Tregs with a donor-specific transgenic TCRs. Such TCR-transduced Tregs have been shown to be potent in the suppression of allograft rejection in a murine model of heterotopic heart transplantation [27]; however, application is limited by mismatch hybridization of the exogenous and endogenous chains [28]. Another approach is to prepare Tregs from antigen-specific normal Foxp3- CD4 T cells by transducing these cells with Foxp3 [29]. With this approach, induction of type 1 diabetes in NOD-scid mice was prevented by prior injection of Foxp3-transduced islet-specific CD4 cells. Such protection was seen with islet-specific BDC cells, though, interestingly, not with GAD-specific cells. Notably, protection against disease correlated with the homing properties of the transferred cells. Thus, BDC cells homed well to the pancreas, whereas GAD cells did not. The clinical potential of this elegant approach has yet to be assessed.

A new and essentially different approach is to induce antigen-specific suppression by cells engineered to express a CAR. These CAR Tregs are normal Tregs transduced to express a hybrid TCR in which the extracellular portion of the receptor is replaced with a single-chain variable region fragment (scFv) of a BCR with known antigen specificity, e.g., binding specificity for HLA-A2. CAR-Tregs recognize native proteins rather than MHC-associated peptides and may be less dependent on IL-2 for their function than conventional Tregs [30]. Notably, CAR-Tregs have been shown to be highly effective at enhancing allograft survival in preclinical models [31,32,33]. Thus, in mice, adoptive transfer of nTregs expressing an HLA-A2-specific CAR can prevent or greatly prolong survival of HLA-2+ heart allografts and human skin xenografts and also protect against GvHD [31].

In summary, the engineering of antigen-specific Tregs by transduction of TCRs or CARs has been successful in preclinical studies, both in vitro and in vivo, and the first clinical trials utilizing CAR Tregs are already recruiting patients (32). Tregs generated by these engineering approaches are likely to be superior to polyclonal Tregs in terms of feasibility and efficacy, although direct evidence on this important issue will have to await the results of clinical trials.

### 3.2. Personalized Immunosuppression—Realistic Goals of Treg Therapy?

Despite the overall success of organ transplantation, the life-long need for immunosuppressive therapy remains a significant problem and contributes to the increased morbidity and mortality seen among transplant patients. Treg therapy is exceptionally potent in controlling acute and chronic rejection in murine models but as mentioned above, is generally unable to lead to a permanent state of tolerance to the graft in the absence of immunosuppression [34]. Permanent tolerance was found in studies with islet allografts [35] but has not been seen with heart or skin allografts [27,36]. With polyclonal Tregs, successful tolerance might be achieved by boosting either the numbers or survival of the transferred cells. This approach could be dangerous, however, because prolonged presence of Tregs in large numbers may predispose to infection. This problem might be circumvented by selectively boosting the survival of antigen-specific TCR-transduced Tregs, or CAR-Tregs, without affecting numbers of normal Tregs. Technically this would be difficult, but might perhaps be achieved by engineering the cells to have elevated sensitivity to endogenous cytokines, e.g., by cotransduction of CAR-Tregs with receptors designed to have increased binding affinity for IL-2 or IL-7; physiological contact with these cytokines could then lead to long-term elevated survival of the donor Tregs but not affect endogenous Tregs.

Currently, however, clinical studies on Treg therapy will most likely focus on the lesser goal of utilizing the capacity of Tregs to reduce the need for continuous immunosuppression. On this point, it should be mentioned that many of the immunosuppressive drugs used for the prevention of allograft rejection induce negative effects on Treg cell numbers and/or function, thereby directly interfering with the efficacy of autologous Treg therapy. Here, calcineurin inhibitors (CNIs), which are the backbone of most immunosuppressive regimens, have been shown to lead to decreased circulating Treg numbers in kidney and liver transplant patients [37,38]. Like Cyclosporin A and Tacrolimus, CNIs act by inhibiting the translocation of NFAT into the nucleus, thereby blocking Teff function and synthesis of IL-2, the latter being crucial for survival and maintenance of Tregs, though less important for Teff [39,40]. Importantly, low-dose IL-2 therapy has been shown to overcome CNI-induced Treg dysfunction and lead to their expansion, suggesting a synergistic interaction between the effects of CNI and IL-2 for the prevention of allograft rejection [41].

For other immunosuppressive drugs, the costimulation blocker CTLA4Ig (belatacept) was approved for use in renal transplantation after it was shown to be equally as effective as CNIs. Significantly, despite a higher incidence of acute rejection episodes, patients treated with CTLA4Ig showed superior renal function compared to CNI-treated patients [42] while displaying a comparable reduction in the frequency and function of Tregs, as also shown in mouse models [43,44,45]. In mice, allograft survival induced by CTLA4Ig was Treg-dependent, though only with suboptimal (low) doses of CTLA4Ig [44]. In line with this finding, in vivo Treg expansion has been shown to improve the efficacy of CTLA4Ig [46]. These findings imply that at low doses, CTLA4Ig can act synergistically with Tregs. At chronic high doses, however, CTLA4Ig ablates Treg frequency, presumably via B7 (CD80, CD86) blockade. In this respect, the survival of Tregs is known to be strongly dependent on CD28/B7 interaction [45]. Methylation of CpG islands within the TSDR regions is suggested to be responsible for impaired Treg function following long-term costimulation blockade with belatacept [47,48].

The finding that the immunosuppressive drugs used for maintenance therapy in transplantation have negative effects on Treg cell numbers and function is clearly troubling. Hence, for Treg therapy, there is increasing interest in developing methods to protect Tregs from the deleterious effects of immunosuppressant drugs. Here, there are ongoing attempts to enhance Treg numbers and function in vivo by co-treatment with various compounds, including rapamycin and HDAC inhibitors and by complement receptor blockade [43]. The benefits of these approaches are still unclear.

### 3.3. Clinical Tolerance: The Criteria for Withdrawing Immunosuppression

One of the biggest hurdles in the design of clinical tolerance trials is the lack of reliable biomarkers and functional assays for identifying tolerant patients. In particular, there are only vague guidelines for reducing or withdrawing immunosuppression. Nevertheless, useful insights on how to define operational tolerance have come from case reports of immunosuppression withdrawal, initiated either because of life-threatening side effects of immunosuppression or, more usually, nonadherence by the patient. For most patients, immunosuppression withdrawal results in acute cellular rejection of the graft, requiring reinstitution of immunosuppression. Some patients, however, develop a state of “operational tolerance” in which the immune response against the graft is abrogated while the immune response against infections and other foreign antigens is preserved. Nevertheless, spontaneous operational tolerance is a rare event, occurring mostly in liver transplant patients with a frequency of around 20% (between 5.6% and 62.6% [49]). For kidney transplantations, the estimated incidence of tolerance is very low, between 0.03% and 5% [50]. Furthermore, in contrast to liver recipients, weaning kidney transplant patients off immunosuppression risks irreversible rejection and graft loss [51,52]. The definition of “operational tolerance” also differs between reports and usually signifies “stable organ function” for a limited amount of time [50], thereby creating a gap between experimental and clinical definitions of tolerance. Thus, patients experiencing subclinical rejection may meet the criteria for operational tolerance, even though biopsy results would likely have revealed early-stage immune injuries. Hence, without biopsy, operational tolerance is highly conjectural. In preclinical animal models, by contrast, tolerance criteria are stringent [53]. Thus, tolerance can be verified by selective survival of a second donor-derived graft (usually skin) as well as by histology results and donor-specific hyporesponsiveness in vitro. In the clinical setting, however, there is still a lack of meaningful assays for defining tolerance at a cellular level. Indeed, the questionable relevance of current functional in vitro assays and the restricted availability of appropriate donor and host cells and tissue for analysis are both seemingly intractable problems.

Despite these problems, there is currently much interest in the use of accessible biomarkers for gauging operational tolerance, either by investigating whole-blood gene expression signatures, flow-cytometry-based immune phenotyping, or both. Interestingly, although tolerance involves multiple host cell types, the relative contribution of these calls to tolerance appears to vary according to the allograft in question. For example, in addition to T cells, changes in host B cells can signify tolerance for kidney transplants [50,54], whereas alterations in NK and γδT cells may indicate tolerance for liver transplants [55,56]. For normal T cells, biomarkers of tolerance include prolonged expression of exhaustion markers on effector T subsets, and as mentioned earlier, elevated numbers of donor-specific Tregs [57]. However, which—if any—of these various noninvasive biomarkers are indicative of stable tolerance is still far from clear. Hence, currently, the decision to withdraw transplant patients from immunosuppressive drugs relies mainly on the physician’s experience, accompanied by the results of multiple tests for tolerance to the graft. There is a pressing need for further studies to validate biomarkers in order to establish organ-specific guidelines for weaning protocols.

## 4. Future Strategies

Despite the success of therapy with genetically engineered Tregs in animal models, clinical use of these cells is still in its infancy. Preclinical data suggest that widely accepted current protocols relying on administration of Tregs alone are not satisfactory for the induction of stable donor-specific tolerance in immunocompetent wild-type animals. As discussed below, CAR Tregs are of particular interest though designing and preparing these cells for use in humans is a major challenge.

### 4.1. Clinical Use of CAR Tregs

As mentioned earlier, redirection of Treg specificity by the use of synthetic CARs has demonstrated therapeutic potency in several preclinical studies [58]. For clinical application, nonhuman primate (NHP) models are an important step for translation to transplant patients and crucially can provide tissue samples that are impossible to obtain from human studies. Recently, Ellis et al. [59] have been able to optimize NHP CAR Treg manufacturing, thereby paving the way for optimization of timing, dosage, induction and maintenance immunosuppression for clinical trials. The first-in-human clinical trials using alloantigen-specific CAR Tregs in kidney and liver transplant recipients have already started (Table 1; NCT04817774, NCT05234190) and are designed to evaluate the safety and feasibility of HLA-A2-specific CAR Treg in clinical kidney transplantation. These studies will provide invaluable information on safety issues and therapeutic potential, thus paving the way for moving to broader clinical applications. In addition, further studies with preclinical models are needed to increase the potency of CAR Tregs, e.g., by engineering the cells to display greater stability and suppressive capacity in vivo and avoid the onset of exhaustion.

### 4.2. In Vivo Approaches for Expanding Tregs: IL-2 Complexes and IL-2 Engineering

Expanding normal and genetically engineered Tregs in vitro before transfer is costly and time-consuming, as well as being technically demanding. For these reasons, stimulating Tregs to expand under physiological conditions in vivo is an attractive alternative (Figure 3). Here, repeated injection of low doses of IL-2 has been shown to augment numbers of circulating murine and human Tregs (59); such expansion is selective for Tregs and reflects the high density of IL-2R on these cells (see below). Low-dose IL-2 treatment stimulates all Tregs, irrespective of their specificity, though it does not promote migration to the organ allograft. Nevertheless, low-dose IL-2 treatment has emerged as a promising method for tolerance induction, both in rodent models and in clinical trials [60]. A concern, however, is that this procedure can be associated with an increased risk of T-cell-mediated rejection [24]. Thus, in a liver transplantation trial, low-dose IL-2 was shown to elicit an IFNγ-dependent inflammatory response within the allograft. This finding presumably reflects that in addition to expanding Tregs, low-dose IL-2 treatment also leads to concomitant proliferation of CD8 T cells and NK cells, though at a lower level than for Tregs.

Another problem with low-dose IL-2 therapy is that the half life of IL-2 is very short, hence IL-2 has to be administered by repeated infusion. However, the half life of IL-2 and other related cytokines in vivo can be greatly extended by complexing the cytokines with specific antibodies before injection [61]. Moreover, in addition to increasing the duration and magnitude of T-cell responses in vivo, cytokine/antibody complexes can be used to selectively stimulate particular T-cell subsets. Thus, complexing IL-2 with different monoclonal anti-IL-2 antibodies (mab) before injection showed that IL-2/IL-2 mab complexes can be used to selectively stimulate proliferation of either NK and CD8 cells or Tregs [62,63]. For anti-IL-2 mAb JES6-1, injecting mice with preformed IL-2/JES6-1 complexes induced the selective elevation of Treg levels in vivo and prevented the rejection of fully MHC-mismatched pancreatic islet allografts without immunosuppression [35]. Furthermore, we recently showed that IL-2/JES6-1 complexes synergize with short-term IL-6 blockade and rapamycin to induce a marked increase in the survival of fully MHC-mismatched skin allografts [36]. Here, it is of interest that even after eventual graft rejection, there was no evidence of sensitization to the graft antigens, as indicated by the normal mixed-lymphocyte reactions and the absence of accelerated rejection of a second donor skin graft. In addition, the mice failed to develop antidonor IgG antibodies, indicating that IL-2 complex treatment blocked both cellular and humoral immunity. Collectively, the data suggest that the immunosuppression mediated by the IL-2/mab complexes led to a prolonged state of immune “ignorance”, followed by gradual restoration of naïve immunity to the graft antigens.

For preferential stimulation of Tregs, JES6-1 and related mab appear to bind to epitopes on IL-2 that guide the IL-2/mab complexes to selectively stimulate cells that express the high-affinity IL-2Rαβγ receptor, i.e., Tregs, but not typical CD8 cells and NK cells. Recently, a fully human anti-IL-2 mab (F5111.2) for selective in vivo Treg expansion was developed [64]. When complexed to human IL-2, this mab led to preferential STAT5 phosphorylation of Tregs in vitro and selective expansion of Tregs in vivo. Functionally, the IL-2/mab complexes displayed potent suppressive activity as measured in murine models of diabetes, EAE and xenogeneic GVHD [64]. More recently, another IL-2 mab has been identified that selectively activates Tregs in mice, NHPs and humans [65] and the first clinical trials are eagerly awaited.

An alternative approach for extending the half life and specificity of IL-2 is to use genetic engineering to prepare IL-2 “superkines” or “muteins” that bind selectively to certain components of the IL-2R. Here, extending studies with the S4B6 IL-2 mab that blocks IL-2 binding to IL-2Rα (CD25) [66], IL-2 has been mutated to bind poorly to the α-chain of the high-affinity IL-2Rαβγ present on Tregs but retain binding for IL-2Rβγ, i.e., the low-affinity IL-2R found on CD8 and NK cells. Hence, like IL-2/S4B6 complexes, IL-2 muteins with poor binding to Tregs can be used to selectively stimulate CD8 cells and NK cells, thereby being useful for cancer immunotherapy [67,68]. Extending this approach, muteins prepared with reciprocal strong binding to IL-2Rα but limited affinity for IL-2Rβγ can be used for selective Treg expansion. When attached to IgG to increase the half life in vivo, these muteins have been shown to prevent spontaneous diabetes in NOD mice [69]. In ongoing clinical trials, the safety and efficacy of a human IL-2 mutein Fc fusion protein (Efavaleukin Alfa) to selectively stimulate Tregs is being evaluated in patients with GvHD (NCT03422627) and systemic lupus erythematosus (NCT03451422).

### 4.3. Treg Cell-Derived Extracellular Vesicles/Exosomes

Extracellular vesicles (EVs) are membrane structures of diverse origin, size and cargo that play a critical role in intercellular communication [70]. Exosomes are a subset of EVs generated via inward budding of former endosomes and are released from cells after fusion with the plasma membrane. Since exosomes carry a spectrum of cell components, including specific receptors, nucleic acids, and MHC molecules, absorption of exosomes by other cells can alter their function. Indeed, uptake of graft-derived exosomes by host APC can play an important role in initiating graft rejection [71,72].

Significantly, Treg-derived exosomes display immunomodulatory and tolerogenic properties [73]. Thus, Smyth et al. demonstrated that the release of Treg-derived exosomes inhibited in vitro T-cell proliferation via a CD73-dependent mechanism [74]. Likewise, another study reported prevention of colitis and systemic inflammation by inhibitory microRNAs (miRNAs) derived from Treg exosomes. Such miRNAs are thought to directly target Th1-cell-mediated responses and thereby guide specific Treg-cell-mediated function to prevent pathogenic inflammation [75]. Here, in rat models of kidney transplantation, the use of adoptive Treg-based exosome transfer resulted in prolongation of allograft survival [76,77]. Similarly, in a rat liver transplantation model, Treg-derived exosomes were shown to suppress CD8 cytotoxicity and prolong survival of liver allografts [78].

These interesting preclinical studies have demonstrated the feasibility and safety of exosome therapy and raise the possibility that this approach could also be used to treat autoimmune disease. On this point, Treg-derived exosomes from patients with remitting multiple sclerosis (MS) were found to be dysfunctional and displayed diminished suppressive capacity [79], suggesting that MS could be treated by transfer of exosomes from normal Tregs. A problem here, however, is that being derived in part from the plasma membrane, exosomes from heterologous Tregs would could lead to donor sensitization, making repeated treatment unfeasible. Hence, unless exosomes could be prepared from functionally-normal autologous Tregs, it is difficult to foresee a use for exosome therapy for autoimmune diseases. Although exosomes are insufficient to activate T cells in vitro in the absence of APCs, allogeneic exosomes can be captured by APCs, leading to so-called MHC cross-dressing [71].

For transplant patients, rejection of infused exosomes is presumably not an issue because exosomes are prepared from autologous rather than heterologous Tregs. However, despite success in animal models, suppression by Treg exosomes appears to be less efficient than contact with intact Tregs [75]. Hence, in the short term, Treg therapy is likely to focus more on transfer of intact Tregs rather than on exosomes from these cells. In the future, however, further advances in Treg engineering might substantially improve the efficacy of exosomes for tolerance induction. Hence, the potential of exosome therapy will continue to elicit interest.

## 5. Conclusions

Though still largely an experimental procedure, Treg adoptive cell therapy is now a clinical reality. Results from clinical trials are encouraging, but many questions remain to be addressed before this approach becomes routinely applicable to transplant recipients. The first systematic evaluation of Treg therapy in a transplant setting was accomplished in the ONE study by a large multicentric team of specialists for kidney transplantation. However, the ONE study was a nonrandomized trial and was not designed to show superiority (or even noninferiority) over standard therapy. So far, data on efficacy in terms of tolerance induction and successful weaning from immunosuppression are restricted to a single study on liver transplantation. Results from ongoing trials and data on long term outcome will hopefully provide crucial additional insights on the potency of Treg therapy in transplant patients and are eagerly awaited. Furthermore, conducting standardized trials using rationally designed Treg-friendly immunosuppressive regimens will be essential for comparing different Treg therapeutics for their efficacy.

In conclusion, although current data suggest that Treg therapy alone might be insufficient for the induction of full immune tolerance in transplantation, there is now optimism that Treg therapy will eventually become a valuable method for substantially reducing the need for continuous immunosuppression in transplant patients.

## Figures and Tables

**Figure 1 ijms-24-01752-f001:**
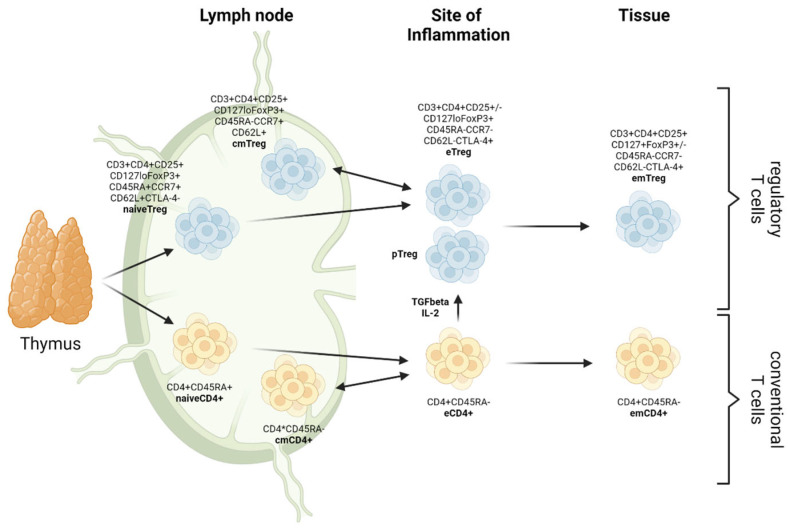
Differentiation dynamics and plasticity of (regulatory) T cells. cm: central memory; em: effector memory; e: effector; p: peripheral (created with biorender.com).

**Figure 2 ijms-24-01752-f002:**
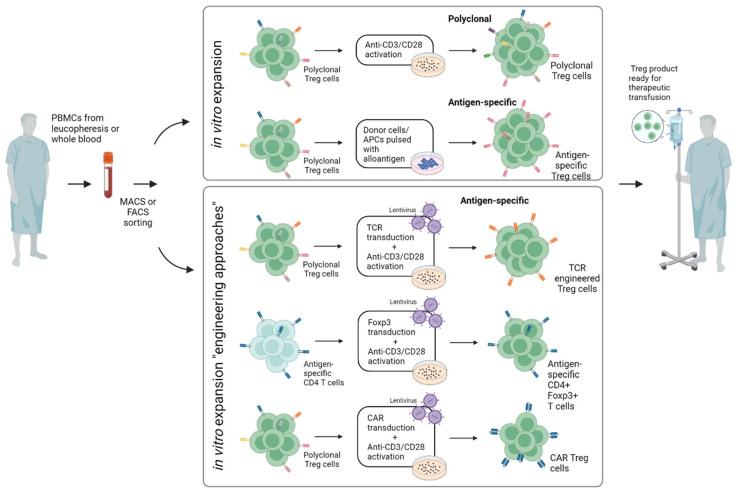
Different approaches for in vitro expansion of Tregs. Engineering approaches are supposed to increase specificity and feasibility. PBMCs: peripheral blood mononuclear cells; FACS: fluorescence-activated cell sorting; MACS: magnetic-activated cell sorting; APC: antigen-presenting cell; TCR: T cell receptor; CAR: chimeric antigen receptor (created with biorender.com).

**Figure 3 ijms-24-01752-f003:**
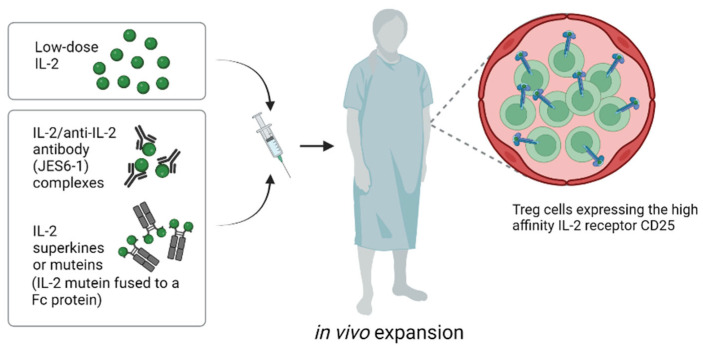
Approaches for in vivo expansion of Tregs (created with biorender.com).

## Data Availability

Not applicable.

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
