# Peer review of "Treg Therapy for the Induction of Immune Tolerance in Transplantation—Not Lost in Translation?"

_ijms, 2023, doi:10.3390/ijms24021752_

Round 1
Reviewer 1 Report
This manuscript presents a comprehensive review of treatment with regulatory T cells (Tregs) in order to achieve a status of tolerance towards the graft in patients after a solid organ transplant.
· In the introduction, the authors present an excellent overview of various types of Tregs.
· The subsequent description of clinical trials refers where possible to data in preclinical models. Clinical trials (status September 2022) are summarized in a table which provides an excellent overview of Treg in various forms of solid organ transplantation. Individual trials are described in more detail, focusing on the way of Treg preparation. In general, Treg therapy was found to be feasible and safe. In a few trials there were side effects, mainly chronic rejection. Some trials were encouraging regarding efficacy, defined by a reduction in chronic immunosuppression. Complete elimination of immunosuppression was reported for only one study.
· In a section on “Open Questions” the authors point to the fact that there is no unequivocal conclusion on safety, which in part is related to differences in methods used in preparation of Tregs The authors point to the possibility that Tregs generated by engineering approaches to increase their specificity might be more potent. They also discuss the balance between dose of Tregs and the emergence of infection, the balance between Treg function and chronic maintenance immunosuppression, and the potential synergism with CTLA4Ig. Also, as the final aim of Treg therapy is achieving tolerance like in animal models, complications in the definition and assessment of tolerance are presented, even regarding the use of the designation “operational tolerance”: essentially there are no validated biomarkers for tolerance.
· In a final section on future strategies the authors discuss chimeric antigen receptor (CAR) T regulatory cells, mentioning some clinical studies that have been initiated. Also, options to expand Treg in vitro, or, as a more preferred alternative, in vivo using interleukin 2 or derivatives are discussed; a recently initiated clinical trial with human IL-2 mutein Fc fusion protein (Efavaleukin alfa) is mentioned in this respect. Also, the option of Treg cell-derived extracellular vesicles/exosomes is presented, with however low expectations regarding efficacy.
· The authors conclude that Treg therapy has grown from encouraging unequivocal results regarding tolerance induction in animal models to a reality in clinical trials. A main conclusion from clinical explorations is that Treg therapy alone might be insufficient for the induction of full immune tolerance in transplantation. Present perspectives are rather that Treg therapy will result in reduced need for immunosuppression in transplant patients. Noteworthy, the authors point to the lack of properly designed randomized clinical trials needed to show superiority, or even non-inferiority in comparison with standard therapy.
This manuscript is an excellent comprehensive review presenting not only the positive aspects and perspectives but also limitations and concerns that need to be addressed in future work. There are only a few suggestions to increase readability:
· Table 1 is difficult to read and needs editing. It is expected that presentation in horizontal rather than portrait format will result in increased readability.
· There many abbreviations which are not commonly used. It is advised to include a list of abbreviations, and also use abbreviations consistently. An example is the abbreviation SOT.
Author Response
Response to REVIEWER 1 comments
This manuscript presents a comprehensive review of treatment with regulatory T cells (Tregs) in order to achieve a status of tolerance towards the graft in patients after a solid organ transplant.
In the introduction, the authors present an excellent overview of various types of Tregs.
The subsequent description of clinical trials refers where possible to data in preclinical models. Clinical trials (status September 2022) are summarized in a table which provides an excellent overview of Treg in various forms of solid organ transplantation. Individual trials are described in more detail, focusing on the way of Treg preparation. In general, Treg therapy was found to be feasible and safe. In a few trials there were side effects, mainly chronic rejection. Some trials were encouraging regarding efficacy, defined by a reduction in chronic immunosuppression. Complete elimination of immunosuppression was reported for only one study.
In a section on “Open Questions” the authors point to the fact that there is no unequivocal conclusion on safety, which in part is related to differences in methods used in preparation of Tregs The authors point to the possibility that Tregs generated by engineering approaches to increase their specificity might be more potent. They also discuss the balance between dose of Tregs and the emergence of infection, the balance between Treg function and chronic maintenance immunosuppression, and the potential synergism with CTLA4Ig. Also, as the final aim of Treg therapy is achieving tolerance like in animal models, complications in the definition and assessment of tolerance are presented, even regarding the use of the designation “operational tolerance”: essentially there are no validated biomarkers for tolerance.
In a final section on future strategies the authors discuss chimeric antigen receptor (CAR) T regulatory cells, mentioning some clinical studies that have been initiated. Also, options to expand Treg in vitro, or, as a more preferred alternative, in vivo using interleukin 2 or derivatives are discussed; a recently initiated clinical trial with human IL-2 mutein Fc fusion protein (Efavaleukin alfa) is mentioned in this respect. Also, the option of Treg cell-derived extracellular vesicles/exosomes is presented, with however low expectations regarding efficacy.
The authors conclude that Treg therapy has grown from encouraging unequivocal results regarding tolerance induction in animal models to a reality in clinical trials. A main conclusion from clinical explorations is that Treg therapy alone might be insufficient for the induction of full immune tolerance in transplantation. Present perspectives are rather that Treg therapy will result in reduced need for immunosuppression in transplant patients. Noteworthy, the authors point to the lack of properly designed randomized clinical trials needed to show superiority, or even non-inferiority in comparison with standard therapy.
Response: We want to thank the reviewer for his time and valuable comments.
This manuscript is an excellent comprehensive review presenting not only the positive aspects and perspectives but also limitations and concerns that need to be addressed in future work. There are only a few suggestions to increase readability:
Table 1 is difficult to read and needs editing. It is expected that presentation in horizontal rather than portrait format will result in increased readability.
Response: We prepared an extra file for the table, being displayed in landscape format, We also increased font size for better readability.
There many abbreviations which are not commonly used. It is advised to include a list of abbreviations, and also use abbreviations consistently. An example is the abbreviation SOT.
Response: We included a list of abbrevations in the revised manuscript.

Reviewer 2 Report
This is an up-to-date and comprehensive review of Treg therapy in organ transplantation. The authors have managed to include information from most of the major articles on this topic reporting basic and translational research. They have also included most of the available information on clinical trials with Tregs (in Table 1).
To make the article more helpful to readers, I suggest that the authors include illustrations in each section (e.g., cartoons of flow charts) that outline the key points of each approach and the results in animals or humans.
I have read most of the articles in the reference list and find the information derived from the articles too brief/dense for readers who could use this review to understand the key points of each experimental or clinical therapy. The figures will add much to this work.
Author Response
Response to REVIEWER 2 comments
This is an up-to-date and comprehensive review of Treg therapy in organ transplantation. The authors have managed to include information from most of the major articles on this topic reporting basic and translational research. They have also included most of the available information on clinical trials with Tregs (in Table 1).
Response: We thank the reviewer for his/her/their time and helpful comments and suggestions.
To make the article more helpful to readers, I suggest that the authors include illustrations in each section (e.g., cartoons of flow charts) that outline the key points of each approach and the results in animals or humans.
Response: We added 2 more Figures to summarize in vitro approaches for Treg specificity (NEW FIG.2) and in vivo Treg expansion (NEW FIG.3)
I have read most of the articles in the reference list and find the information derived from the articles too brief/dense for readers who could use this review to understand the key points of each experimental or clinical therapy. The figures will add much to this work.
Response: We appreciate the suggestion and added 2 more figures to summarize key points and approaches for the readers.
